# Gut Dysbiosis and Blood-Brain Barrier Alteration in Hepatic Encephalopathy: From Gut to Brain

**DOI:** 10.3390/biomedicines11051272

**Published:** 2023-04-25

**Authors:** Ali Shahbazi, Ali Sepehrinezhad, Edris Vahdani, Raika Jamali, Monireh Ghasempour, Shirin Massoudian, Sajad Sahab Negah, Fin Stolze Larsen

**Affiliations:** 1Cellular and Molecular Research Center, Iran University of Medical Sciences, Tehran 1449614535, Iran; shahbazi.a@iums.ac.ir (A.S.); sh.massoudian@gmail.com (S.M.); 2Department of Neuroscience, Faculty of Advanced Technologies in Medicine, Iran University of Medical Sciences, Tehran 1449614535, Iran; monireh97@gmail.com; 3Neuroscience Research Center, Mashhad University of Medical Sciences, Mashhad 9919191778, Iran; 4Department of Microbiology, Faculty of Medicine, Mazandaran University of Medical Sciences, Sari 4815733971, Iran; edrisvahdani@gmail.com; 5Research Development Center, Sina Hospital, Tehran University of Medical Sciences, Tehran 1417653761, Iran; 6Digestive Disease Research Institute, Tehran University of Medical Sciences, Tehran 1417653761, Iran; 7Department of Neuroscience, Faculty of Medicine, Mashhad University of Medical Sciences, Mashhad 9919191778, Iran; 8Shefa Neuroscience Research Center, Khatam Alanbia Hospital, Tehran 9815733169, Iran; 9Department of Intestinal Failure and Liver Diseases, Rigshospitalet, Inge Lehmanns Vej 5, 2100 Copenhagen, Denmark

**Keywords:** hepatic encephalopathy, gut microbiome, blood-brain barrier, inflammation, gut-liver-brain axis, bacterial metabolites, tight junctions

## Abstract

A common neuropsychiatric complication of advanced liver disease, hepatic encephalopathy (HE), impacts the quality of life and length of hospital stays. There is new evidence that gut microbiota plays a significant role in brain development and cerebral homeostasis. Microbiota metabolites are providing a new avenue of therapeutic options for several neurological-related disorders. For instance, the gut microbiota composition and blood-brain barrier (BBB) integrity are altered in HE in a variety of clinical and experimental studies. Furthermore, probiotics, prebiotics, antibiotics, and fecal microbiota transplantation have been shown to positively affect BBB integrity in disease models that are potentially extendable to HE by targeting gut microbiota. However, the mechanisms that underlie microbiota dysbiosis and its effects on the BBB are still unclear in HE. To this end, the aim of this review was to summarize the clinical and experimental evidence of gut dysbiosis and BBB disruption in HE and a possible mechanism.

## 1. Introduction

Hepatic encephalopathy (HE) is the main complication of acute liver failure (ALF) and advanced liver diseases; it is characterized as a set of complex spectra of brain dysfunctions and neuropsychiatric impairments, ranging between preclinical changes and deep coma [1,2]. In addition to economical burdens, HE severely affects daily routines and reduces the quality of life in patients and their caregivers [3]. According to the etiology of the disease, HE is divided into three types. Type A is associated with ALF, type B is associated with portosystemic shunts, and type C results from portal hypertension and cirrhosis. The European Association for the Study of the Liver (EASL) has categorized HE as being either covert HE and overt HE, according to the severity of the complications. Based on the West Haven Criteria, covert HE is concluded as minimal or grade I of HE, which is only diagnosed by neuropsychiatric tests, while overt HE is considered grade II-IV of HE [4]. The prevalence of HE is associated with the etiology of the disease, the severity of the complications, and the type of HE. The prevalence of covert or minimal HE is about 20–80% in patients with cirrhosis [5,6,7,8,9]. Approximately 30–40% of patients with cirrhosis experience overt HE during their course of the disease and it is associated with risk factors such as ascites, infections, esophageal bleeding, diabetes, and covert HE [10,11,12,13]. Although the pathogenesis of HE remains incompletely elucidated, the central role of accumulated gut-derived agents, particularly ammonia, has been confirmed [14]. An axial role for gut microbiota in the pathophysiology of HE, which known as the gut-brain axis, has also been suggested [15]. Furthermore, the impairment of the blood-brain barrier (BBB) has been reported in many experimental models, as well as in patients with HE. This review aims to demonstrate how gut dysbiosis disrupts the integrity of the BBB following HE and to discuss how alterations in gut microbiota diversity affect the BBB integrity in HE.

## 2. Gut Dysbiosis in Hepatic Encephalopathy

The term gut microbiome, previously known as gut normal flora, encompasses all microorganisms, including bacteria, fungi (fungal mycobiome), viruses (virome), and archaea that inhabit the small intestine and colon [16]. One to two years after birth, the microbiota community is created, and its diversity is altered under the influence of age, diet, lifestyle, and geography during one’s lifetime [17]. The gut microbiomes synthetize the enzymes that break down some indigestible polysaccharides and carbohydrates to absorbable glucose and short-chain fatty acids. The gut microbiomes also are responsible for the production of vitamins, including vitamin K, cobalamin (B_12_), biotin (B_7_), and folate (B_9_), as well as facilitating the absorption of calcium, magnesium, and iron by enterocytes [18]. Normal gut microorganisms compete for luminal contents with pathogens and prevent the colonization of these organisms within the gut. Furthermore, gut microbiomes are in close contact with the lamina propria where a huge number of intestinal immune cells are located that develop the host immune system and increase intestinal protection [19]. The intestinal tract is well-innervated by the enteric nervous system and vagus nerve, which comprise the transmission pathways for signals from the gut to the central nervous system (CNS) [20]. Some clinical and experimental evidence has confirmed that gut dysbiosis happens in cirrhotic patients with HE and experimental animal models (Table 1). These studies are mainly conducted through metagenomics and metatranscriptomics approaches (Box 1). Gut microbiota analysis in cirrhotic patients with HE revealed a significant decrease in Clostridiales XIV, Ruminococcaceae, and Lachnospiraceae as the autochthonous taxa with a significant enhancement in Enterococcaeae, Staphylococcaceae and Enterobacteriaceae as the pathogenic taxa compared to healthy controls [21]. Increased pathogenic taxa (i.e., Enterobacteriaceae and Enterococcaceae) and decreased autochthonous taxa (i.e., Lachnospiraceae, Ruminococcaceae and Clostridiales XIV) were also reported in fecal analysis in cirrhotic patients with HE compared to cirrhotic patients without HE [22]. In addition, dysbiosis was well correlated with systemic inflammation in these patients [22].

Box 1Metagenomics and metatranscriptomics analyses in microbiome studies.Studies in microbiota communities are a field of great interest due to changes in their composition associated with health and diseases. To identify the structure and functional dynamics of the gut microbiome, several multi-omics approaches such as metagenomics (DNAs interactions) and metatranscriptomics (RNAs interactions) are applicable. Each approach images different types of biological information from the microbiota communities. Current investigations combine more than one omics approach to image a high-resolution picture of the gut microbiota dynamics [41]. Genomics is concerned with identifying the genetic composition of a single bacterium, but metagenomics is concerned with identifying the genetic composition of an entire community of bacterial communities (for instance, the entire gut microbiome of humans) [42]. Metagenomics consists of several culture-independent techniques (i.e., experimental and bioinformatic approaches) to analyze extracted DNA directly from biological samples, such as saliva, stool, and sputum [43]. In summary, the DNA extracted from bacterial communities amplified the DNA containing 16S ribosomal RNA (rRNA as a housekeeping genetic marker) gene and was sequenced. Afterwards, similar sequences as operational taxonomic units (OTUs) were compared to several reference 16S databases (i.e., RDP, SILVA, and Greengenes) to identify the microbiota composition in the samples based on the OTUs similarities [44,45]. In shotgun metagenomic sequencing, all extracted community DNA is sequenced and compared to some reference genomes in several databases (i.e., KEGG, and BLAST) to specify the abundance, diversity, and function of the microbiome [45,46]. Metatranscriptomics is a powerful tool to specify the expression patterns of genes (functional profile; total mRNA) in sequenced genomes from communities of bacteria in a particular sample. This approach provides information about the diversity of active genes and also identifies differences in the gene expression patterns following health and disease in communities of bacteria [47]. In metatranscriptomics microbiome studies, first the total RNA is extracted from bacterial communities, followed by the purification of the desired RNA (i.e., mRNA, microRNA, and lincRNA) and the ribosomal RNA is removed. Then, the RNA is reverse transcribed into cDNA and is amplified to induce a cDNA library. After that, the library is sequenced for further comparing with reference genomes in some databases (i.e., KEGG, CARD, VFDB, and Kraken) [48,49]

## 3. Blood-Brain Barrier Structure and Transportation Systems

The BBB is a complex and highly selective border that forms an important part of the neurovascular unit in the CNS. This unique structure protects neuronal cells from direct contact with circulatory neurotoxic agents, pathogens, and peripheral inflammation [50]. The BBB is well organized in such a way that allows the passage of some small hydrophobic molecules (i.e., oxygen and carbon monoxide) by passive diffusion, and glucose and amino acids through active transport. The barrier is comprised of one single layer of muscle-loss endothelial cells that are covered by a basement membrane, pericytes, and astrocyte end-feet (Figure 1) [51]. The first obstacle against circulatory neurotoxic molecules is a thin monolayer of endothelial cells that are connected via tight junctions and adherent junctions. The structure of these cells is such that it distinguishes them from endothelial cells in the other tissues. They strongly seal the paracellular pathway through highly expressed tight junction proteins that allow them to severely regulate the flux of ions and molecules between the blood and brain tissue [52]. Cerebral endothelial cells also have lower caveolae on their luminal membrane, a higher number of mitochondria, lower levels of leukocyte adhesion molecule, and a restricted transcytosis rate compared to the endothelial cells in other tissues [52]. The tight junctions in place of endothelial junctional clefts are formed by many transmembrane and cytoplasmic proteins that are responsible for sealing the paracellular pathway and restricting the movement of solutes. Two major transmembrane proteins occludin and claudin (claudin-5), as well as one cytoplasmic protein zonula occludens-1 (ZO-1), are formed in the endothelial tight junctions (Figure 1) [53]. Occludin and claudin are integral membrane proteins that have a significant role in the sealing and barrier function of the endothelial junctional clefts, while ZO-1, as a membrane-associated guanylate kinase and a scaffold protein, anchors transmembrane proteins to the actin cytoskeleton, which intensifies the stability of the tight junctions and has a significant role in signaling between cells [53]. Another group of proteins, adherens proteins, are expressed in the junctional clefts that stabilize and regulate the function of tight junction transmembrane proteins. Some important adherens proteins that are present in the endothelial junctional clefts include junctional adhesion molecules (JAMs), platelet endothelial cell adhesion molecule 1 (PECAM-1), endothelial cell-selective adhesion molecule (ESAM), and catenins [54].

Only three transmission routes can be considered for solute transportation across the BBB (Figure 1); The first transmission route is a sealed paracellular pathway by tight junction that is allowed hydrated ions to pass in a limited way [55]. The second path is transcytosis transport (specifically endocytosis), which mediates the transportation of some macromolecules and drugs across the BBB. Transcytosis is most commonly observed in two forms through the BBB, such as receptor-mediated transport and absorptive-mediated transport [55,56]. Receptor-mediated transport or clathrin-mediated endocytosis is a transportation pathway that is mediated by clathrin proteins and requires the interaction of a specific substrate to its receptor on the endothelial membrane [57]. Ligand-receptor interaction on the plasma membrane triggers the invagination of the membrane through a clathrin-dependent mechanism [58]. Finally, a ligand in a clathrin-coated pit can enter the cell. Transporting low-density lipoprotein (LDL) by the LDL receptor, iron via the transferrin receptor, and the insulin-like growth factor receptor are three important examples of receptor-mediated transport through the BBB [55,59]. Absorptive-mediated transport or caveolae-mediated transport is another form of transportation for macromolecules and drugs across the BBB that are triggered by the shear stress that is induced by the substrate and its interaction with endothelial glycocalyx at the luminal side [60,61]. The interaction of the ligand with glycocalyx can trigger the oligomerization of the mechanosensory protein caveolins, caveolar invagination, and endocytosis of the ligand [58]. Beyond the aforementioned routes, the brain endothelial cells express a number of transporters to uptake essential substrates as the third transmission route through the BBB [62,63]. This type of transportation is categorized into three forms: active transporter, carrier-mediated transport, and ion transporters [63]. ATP-binding cassette (ABC) transporters such as P-glycoprotein 1 (P-gp) are common examples of active transport in the BBB that hydrolyze ATP to utilize its energy to efflux many substrates or drug metabolisms across the concentration gradient from the brain tissue [64]. Carrier-mediated transporters are responsible for transporting glucose, specific ions, amino acids, organic anions, cations, and other substances with specific properties that cannot directly pass through the cell membrane [65]. Carrier-mediated transporters may be categorized into three types: uniporter, symporter, and antiporter. Glucose transporter 1 (GLUT1) is an example of uniporter-mediated transport, which is mediated by the movement of glucose down its concentration gradient across the BBB [66]. Monocarboxylate transporter 1 (MCT1) is an example of symporter-mediated transport that passes both the proton and lactate in the same direction [67]. Organic anion transporting polypeptides (OATPs) such OATP1A2 are mainly expressed in the apical membrane, and OATP2B1 is expressed in the basal membrane of brain endothelial cells. These are other examples of carrier-mediated transport across the BBB, which mediate the uptake and efflux of neurosteroids and thyroid hormones [68]. L-type amino acid transporter 1 (LAT1) is a common example of antiporter-mediated transport in the BBB. These transporters regulate the influx of neutral essential amino acids (i.e., histidine, tryptophan, leucine, isoleucine, phenylalanine and tyrosine) into the brain endothelial cell and astrocytes in exchange with the efflux of glutamine [69]. The endothelial cells also highly express many ion transporters, such as Na^+^-K^+^ ATPase, Na^+^-K^+^-Cl^−^ cotransporter, K^+^ channels, Na^+^/Ca^2+^, Na^+^/H^+^, and Cl^−^/HCO_3_^−^ exchangers, Na^+^/HCO_3_^−^ symporters and Ca^2+^ transporters severely regulate the sodium gradient concentration to ensure the uptake of sodium-dependent substrates, maintain the intracellular pH, and regulate the brain interstitial ions equilibrium and water content [70].

The second cellular components of the BBB are pericytes, which are located in the basement membrane in close contact with the abluminal side of the endothelial cells via gap junctions and peg-socket junctions [71]. These cells can be visualized by some specific markers, for example, platelet-derived growth factor receptor beta (PDGFRβ) and neuron-glial antigen 2 (NG2) in mice. Pericytes are important in the development of tight junctions and the formation of the BBB. These cells stabilize the brain endothelial cells and protect the brain from the invasion of immune cells [72]. Pericytes also synthetize some proteins that regulate the capillary blood flow and trigger angiogenesis [73]. The paracrine interaction of endothelial cells and pericytes is also well regulated through platelet-derived growth factor beta (PDGF-β), which is released from endothelial cells and binds to its receptors on pericytes. This ligand-receptor interaction triggers signaling cascades that mediate the proliferation and migration of pericytes [74]. Astrocytes are the most numerus glial cells in the CNS that play many important functions, including ions homeostasis, the regulation of the pH in the extracellular space, controlling the cerebral blood flow, providing nutrients to the neurons, CNS repairing following injuries, and the regulation of endothelial cells’ function and neurotransmission [75,76]. The processes of these cells ensheath the microvasculature as astrocyte end-feet that enable them to send signals from activated neurons to regulate the blood flow [77]. The high expression of aquaporin 4 water channels (AQP4) has been identified in astrocytes, indicating that these cells play a major role in water homeostasis and the clearance of waste substances from the brain interstitial space [78]. The dystroglycan–dystrophin complex is also expressed and localized in the astrocyte end-feet that connect the end-feet cytoskeleton to the basement membrane [79,80]. It is concluded that this complex facilitates the localization of AQP4 in the perivascular space to maintain water flow and form a part of the glymphatic system [80]. Therefore, the BBB not only acts as a sealed physical barrier and a transportation border, but also is a metabolic structure, and its constituent cells can affect each other and their vicinity through signaling molecules.

The intact integrity and normal functioning of the cellular components of the BBB ensure the homeostasis of CNS and proper neuronal function [81]. Any small changes in this restricting barrier and its properties can confront the neuronal cells to direct contact with circulatory neurotoxic agents [81,82].

BBB dysfunction and an increase in the CNS levels of neurotoxic substrates has been reported in patients with HE and in experimental models (Table 2). In cases of a disrupted BBB following liver insufficiency, the circulating ammonia, mercaptans, lipopolysaccharide (LPS), and other gut-derived neurotoxic agents pass the BBB and make close contact with the neurons, astrocytes, and microglia, which trigger neuroinflammation and produce reactive oxygen species [83,84,85,86,87]. The disrupted BBB also allows an increase in the entry of immune cells and circulatory pro-inflammatory cytokines to the brain parenchyma and causes neuroinflammation and damage [88,89]. The question is how the BBB is disrupted in HE and whether gut microbiota-derived molecules are implicated.

## 4. Altered Gut Microbial Metabolites and Molecules May Disrupt the Integrity of the BBB in HE

The intestinal microbiome seems to be of central importance in the progression of HE [106,107]. The decreased expression of tight junction proteins and the increased BBB permeabilization have been addressed in the brain of germ-free mice compared with normal gut flora [108]. The gut microbiota produces some molecules and metabolites that can exert beneficial or harmful effects on the host CNS. Short-chain fatty acids (SCFAs, i.e., propionate, butyrate and acetate) are the final product of the bacterial fermentation of non-digestible polysaccharides in the lower intestinal tract and act as signaling molecules, have anti-inflammatory properties, and protect colonic epithelial cells [109]. Some studies have shown that basal non-toxic levels of SCFAs can preserve the intestinal barrier integrity, protect the BBB from oxidative stress, and positively regulate the expression of thigh junction proteins [110,111,112]. The treatment of germ-free mice with sodium butyrate strongly recovered the destructed BBB after visualizing the level of Evans blue in the frontal cortex, striatum, and hippocampus [108]. Moreover, the treatment of rhesus monkeys with antibiotics altered the gut microbiota composition, and in particular, decreased the SCFAs-producing phyla and impaired the permeability of the BBB in the thalamus [113]. A lower amount of propionate, butyrate, and acetate was seen in the fecal samples of cirrhotic patients with HE compared to those without HE [114]. The family Ruminococcaceae is the main source for the production of SCFAs in the human intestinal tract [115]. As shown in Table 1, this SCFAs-producing family of gut bacteria have been decreased in the gut of patients with cirrhosis and HE [22,27]. Specifically, the decreased genera in Ruminococcaceae and Lachnospiraceae families have been reported in fecal samples taken from cirrhotic patients, which were associated with a low capacity to produce butyrate in the intestine of patients [116]. The decreased basal concentration of SCFAs may indirectly affect the integrity of the BBB through the impairment of the intestinal barrier following cirrhosis. A recent systematic review reported that the administration of SCFAs recovers the destructed intestinal barrier and improves the severity of hepatic injury following liver disease [117]. The disruption of the normal physical intestinal barrier may induce microbial and endotoxin translocation, which result in systemic inflammation, liver injury, BBB permeabilization, and neuroinflammation in cirrhosis and HE [118,119]. The intestinal microbiome also contributes to the metabolism of bile acids and mediates the synthesis of a small part of bile acids (as secondary bile acids), which maintain the integrity of the intestinal barrier and regulate the intestinal immune responses through farsenoid X receptor (FXR) and bile acid receptor GPBAR-1 (TGR5) [120,121,122,123]. Decreased concentrations of intestinal secondary bile acids following gut dysbiosis are shown in cirrhosis [124,125]. Studies have shown that the concentrations of circulatory primary bile acids produced by hepatocytes increased [100,105,126,127,128], while the levels of intestinal secondary bile acids produced by the gut microbiome reduced in cirrhosis [124,125]. Therefore, a bile acids imbalance may affect the BBB with two bile acid-dependent approaches in cirrhosis: (1) an increased circulatory concentration of bile acids directly correlated with the BBB permeabilization through a Rac1-dependent mechanism [100]; (2) the decreased intestinal concentration of bile acids indirectly affect the integrity of the BBB via the destruction of the intestinal barrier, bacterial translocation, and the induction of systemic inflammation (Figure 2). The microbe- or pathogen-associated molecular patterns (MAMPs or PAMPs) are other important products of gut microbiota that are identified by expressed pattern recognition receptors (PRRs) on innate immune cells (i.e., natural killer cells, dendritic cells and monocytes/macrophages) [129]. Toll-like receptors (TLRs) are considered as the main groups of PRRs that recognize both MAMPs and endogenous damage-associated molecular pattern molecules (DAMPs) in the intestinal tract [130]. The activation of TLRs leads to a change in the expression of the downstream signaling proteins that mediate the immune responses to initiator agents. Some bacterial endotoxins, such as LPS, flagellin, and bacterial DNA, are considered to be the main MAMPs that are recognized via TLR4, TLR5, and TLR9, respectively, and are responsible for the activation of the immune system and the production of inflammatory cytokines following gut dysbiosis and bacterial infections [131]. A destructed intestinal barrier and intestinal bacterial overgrowth may explain the cause of bacterial translocation in cirrhosis. Clinical and experimental studies have shown bacterial translocation in cirrhosis [132,133,134,135,136,137,138]. The circulatory levels of potent inflammatory MAMPs and LPS increased and were associated with the risk of hospitalization and death in patients with advanced liver disease [139,140]. The activation of TLRs by LPS and other MAMPs can trigger other downstream signaling molecules, such as Interleukin-1 receptor-associated kinases (IRAKs) and TNF receptor-associated factor 6 (TRAF6), which increase the expression of some important inflammatory transcription factors (i.e., nuclear factor Kappa-B, activator protein 1, interferon regulatory factor 3, p38 mitogen-activated protein kinase and c-Jun N-terminal kinase) [129,141,142]. These translocated transcription factors trigger the gene expression of proinflammatory cytokines and chemokines, such as interleukin-1 beta, interleukin-6, tumor necrosis factor alpha, CXCL8 CXC motif chemokine ligand 8, and CXCL10 CXC motif chemokine ligand 10 [143,144,145]. The induction of systemic inflammation via the activation of TLRs in response to MAMPs can activate hepatic kupffer cells, induce liver damage, and make the BBB susceptible to disruption, as seen in experimental studies [83,146,147]. Increased circulatory levels of LPS were correlated with systemic inflammation, the down-regulation of tight junction proteins (i.e., claudin-5 and ZO-1), the production of oxidative stress, and neuroinflammation in rodents [148,149]. It has been reported that circulatory LPS can disrupt the integrity of the BBB via an increase in the expression of brain α-synuclein [150]. In azoxymethane-induced HE mice, the injection of LPS strongly exacerbated the concentration of pro-inflammatory cytokines, worsened hyperammonemia, increased liver injury, up-regulated brain matrix metalloproteinase-9, and severely increased the permeability of the BBB [83]. Moreover, the injection of LPS in galactosamine-induced HE mice led to the shrinkage of the brain endothelial cells, decreased tight junction protein occludin, and the raised extravasation of Evans blue dye into the brain in a TNFα-dependent manner [91]. Other probable gut microbiota-derived products, such as mercaptans and phenol, were able to impair the BBB in acute liver injury [151]. The Kupffer cells are resident liver macrophages that have an important role in the defense against infections and endotoxins under normal conditions [152]. In cirrhosis and liver failure, the impaired intestinal barrier leads to a rise in the circulatory concentrations of gut-derived endotoxins, which can reach the liver and activate Kupffer cells. Activated Kupffer cells trigger cytokine and chemokine production, polymorphonuclear cells infiltration, the induction of oxidative stress, and tissue hypoxia in liver parenchyma, which leads to hepatocyte injury and the release of pro-inflammatory cytokines into the circulation [152,153,154]. Furthermore, shunting ad gut-derived endotoxins through varices will only exacerbate a proinflammatory drive in the systemic circulation and impair the permeability of the BBB by a change in the expression of tight junction proteins, the activation of brain glial cells, and the infiltration of peripheral immune cells [103,105,155,156,157,158,159].

Gut microbiota-derived metabolites also affect the CNS through the enteric nervous system (ENS), which normally connects to the CNS via the vagus nerve and sympathetic pathways [160,161]. Gut microbiota products (derived from the small intestine and large intestinal bacteria; Box 2) can directly regulate the development and function of the ENS through MAMPs or PAMPs-PRRs interactions. The cellular components of ENS (i.e., enteric glial cells and neurons) express several PRRs, such as TLR2, TLR3, TLR4, TLR7, and TLR9, which activate in response to LPS, polysaccharide A, and RNAs, and lead to the regulation of the intestinal motility and contractility, as well as the maintenance of the enteric plexuses [162,163,164]. Furthermore, SCFAs as the final product of the bacterial fermentation of non-digestible polysaccharides, which can activate free fatty acid receptor 3 (FFAR3) on enteric neurons and enteroendocrine cells [165]. SCFAs also regulate the gastrointestinal motility by affecting ENS in rats [166]. In addition to the direct effects of the gut microbiota-derived products on the ENS, these products indirectly regulate the function of ENS through several interface cells (i.e., enterochromaffin cells, enteric macrophages and dendritic cells, and intestinal stromal cells) [167,168,169,170,171,172]. The activation of these intermediate cells by gut microbiota metabolites leads to the production of many products, such as serotonin, neuropeptide Y, gastric inhibitory polypeptide, glucagon-like peptide-1, substance P, bone morphogenetic protein 2, glial cell-derived neurotrophic factor, and transforming growth factor β1, which affect the neurons of the ENS [164,167,173,174,175,176]. Gut dysbiosis following liver diseases may negatively regulate the function of the CNS through the differentially produced gut products that prevent the normal regulatory functions of the ENS that have previously been induced by normal gut microbiome metabolites.

Box 2Small intestine versus large intestine microbiome composition.Bacteria constitute a large part of the gut microbiome compared to other microorganisms (i.e., fungi, viruses, protozoa, and archaea). In addition, the composition of the gut bacteria throughout the gastrointestinal tract (GI) is not the same [177,178]. The communities of bacteria increased from the upper GI (10^4^–10^6^ cells/gram feces) to the lower GI (10^9^–10^12^ cells/gram feces) as a result of the lower pH, rapid movement of luminal contents, and the presence of antibacterial agents in the stomach and duodenum compared to the lower GI [177,179,180]. The family of Lactobacillaceae and Enterobacteriaceae are dominant bacteria in the stomach and small intestine (i.e., duodenum, jejunum, and ileum), while Bacteroidaceae, Prevotellaceae, Rikenellacea, Lachnospiraceae, and Ruminococcaceae are major dominant families of bacteria in the large intestine [179]. In microbiome studies, it is important to note that a new experimental comparative study by Ji-Seon Ahn et al. revealed that the fecal microbiome contents are not acceptable to be generalized to the entire gut microbiome [181].

## 5. Therapeutic Targets

Given that gut dysbiosis may contribute to the pathophysiology of BBB permeabilization in HE, therapeutic interventions that target gut dysbiosis and their metabolites may be useful for restoring the destructed BBB and its consequent immune cells infiltrations, oxidative stress, microglial activation, and neuroinflammation in HE. Antibiotic therapy (mainly rifaximin, neomycin, and metronidazole) targets the gut microbiota and decreases the production and absorption of intestinal ammonia, resulting in alleviated hyperammonemia, decreased hospitalization for overt HE, and an improved quality of life for patients with HE [182,183]. The efficacy of antibiotics on the integrity of the BBB is very limited and only one study has indicated that the administration of rifaximin in hyperammonemic BDL rats significantly decreased the florescent intensity of fluorochrome (an indicator of BBB permeabilization) in the brain tissue [104]. The knowledge of how the alternation and normalization of the gut dysbiosis influence the structure of the BBB in HE remains unknown. However, the modulation of the gut dysbiosis using some therapeutic options, such as probiotics and prebiotics, antibiotic therapy, and fecal microbiota transplantation (FMT), have shown beneficial effects on the BBB integrity in several experimental models, including Parkinson’s disease (AD), Alzheimer’s disease, depression, Gulf war illness, toxicity, inflammation, and aging (Table 3).

Probiotic supplements contain live microorganisms that modulate the composition of the gut microflora, while prebiotics are types of nutrients or agents (i.e., dietary fibers and non-absorbable disaccharide) that are beneficial for the growth and activity of the gut microbiota [199,200]. A combination of probiotic and prebiotic therapy in a chronic mice model of AD prevented striatal neuronal apoptosis, improved the integrity of the BBB, and showed neuroprotection effects [184]. In an APP/PS1 mice model of Alzheimer’s disease, the administration of a combination of beneficial bacteria decreased the signal intensity of Evans blue in hippocampus, increased the tight junction proteins ZO-1 and occludin, and reduced the cerebral concentration of LPS [185]. An enhanced cerebral expression of ZO-1 and occludin was also seen in a rat model of traumatic brain injury after the administration of probiotic Bifidobacterium lactis [201]. The decreased extravasation of Evans blue dye was shown in the prefrontal cortex of mice models of depressive-like behavior after the administration of a new probiotic Komagataella pastoris, KM71H [186]. In the mice model of stress, the administration of Lactobacillus plantarum MTCC 9510 significantly decreased the cerebral levels of Evans blue, increased brain-derived neurotrophic factor (BDNF), and decreased neuroinflammation and cerebral oxidative stress [188]. In addition, the treatment of the senescence-accelerated mouse prone 8 (SAMP8) mouse strain model of aging with some species of Lactobacillus and Bifidobacterium, as well as ProBiotic-4, reduced the BBB damage, decreased the LPS concentrations, and mitigated pro-inflammatory signaling cascades in the brain tissue [190,202]. In mouse models of Gulf war illness as a chronic and multi-symptomatic disorder, the administration of andrographolide decreased intestinal pro-inflammatory cytokines, restored the claudin-5 protein level, increased the BDNF, and decreased the microglial activation in cerebral tissues [191].

FMT is a procedure to collect feces from healthy individuals and transfer them into the recipient’s intestine, mainly in patients with gut dysbiosis [203]. Overall, FMT may improve hyperammonemia through the normalization of the gut microbiome composition to low urease bacteria, increasing the hepatocyte clearance of ammonia, and the improvement of the intestinal barrier structure in liver diseases [204]. The effect of FMT on the structure of the BBB in cirrhosis and HE has been less investigated. In a chronic rotenone administration-induced Parkinson’s disease mouse model, FMT significantly restored the tight junction proteins (i.e., occludin, claudin-5, and ZO-1), decreased the astrocyte reactivity, reduced the microglial activation, decreased the concentration of LPS, and suppressed the inflammatory cascade TLR4/MyD88/NF-κB in substantia nigra [195]. In addition, FMT attenuated the florescent intensity of Evans blue and increased the protein expression of occludin-5 in the spinal cord, as well as reducing the astrocyte reactivity and decreasing the microglial activation in the brain tissue of an experimental autoimmune encephalomyelitis mouse model of multiple sclerosis [196]. In a mice model of spinal cord injury, FMT attenuated the extravasation of Evans blue, increased the expression of ZO-1 and occludin, and decreased astrogliosis and microglial activation in the spinal cord [197].

## 6. Conclusions and Perspective

The gut microbiota contributes to the development and homeostasis of the nervous system. Gut dysbiosis is associated with cognitive impairments, cerebral abnormalities, and other complications in patients with advanced liver diseases. Gut-originated metabolites released from altered pathologic bacteria impair the integrity of BBB, devastate the brain microenvironment, and lead to the development of HE. The modulation of the gut microbiome composition by different agents has been revealed to have beneficial effects in the management of HE. Moreover, the manipulation of the intestinal microbiome to protect the beneficial species and decrease the harmful microbiome can improve and control HE through changes in the levels of circulating neurotoxic metabolites and products. Probiotics and prebiotics, antibiotics, and FMT improve the pathological changes in the structure of the BBB in numerous diseases, such as Alzheimer’s disease, Parkinson’s disease, depression, chronic stress, and stroke. The efficacy of these interventions still remains an open challenge in HE. However, the majority of animal and human studies have only focused on gut bacteria communities and underestimated the mycobiome diversity and virome composition of the gut microbiota in HE. In terms of future perspectives, it may be important to investigate the mycobiome and virome composition of the gut microbiome, along with bacteria, through multi-omics comprehensive metagenomics and metatranscriptomics studies to identify the specific microbial taxa and metabolites that are associated with HE and potentially serve as biomarkers for the diagnosis and monitoring of the disease. In conclusion, the relationship between gut dysbiosis and HE is a complex and dynamic process that requires further investigation. However, the growing body of evidence suggests that targeting gut dysbiosis represents a promising strategy for the prevention and treatment of HE.

## Figures and Tables

**Figure 1 biomedicines-11-01272-f001:**
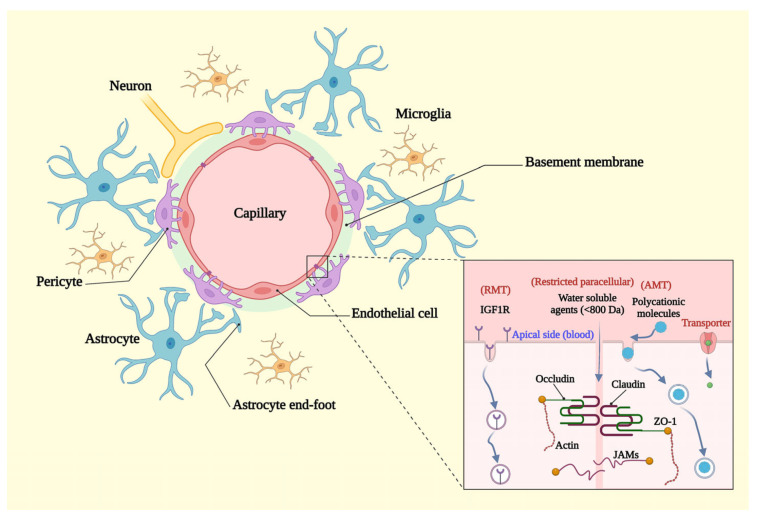
The structure of blood–brain barrier. The BBB is formed by endothelial cells, pericytes, basement membrane and perivascular astrocyte end-feet. This barrier strongly restricts paracellular transmission due to the presence of tight junctions between endothelial cells. Tight junctions include two transmembrane proteins claudin and occludin and an intracellular protein zonula occludens-1. AMT: Absorptive-mediated transport; IGF1R: Insulin-like growth factor1 receptor; JAMs: Junctional adhesion molecules; RMT: Receptor-mediated transport; ZO-1: zonula occludens-1. Created with “BioRender.com. (accessed on 2 February 2023)”.

**Figure 2 biomedicines-11-01272-f002:**
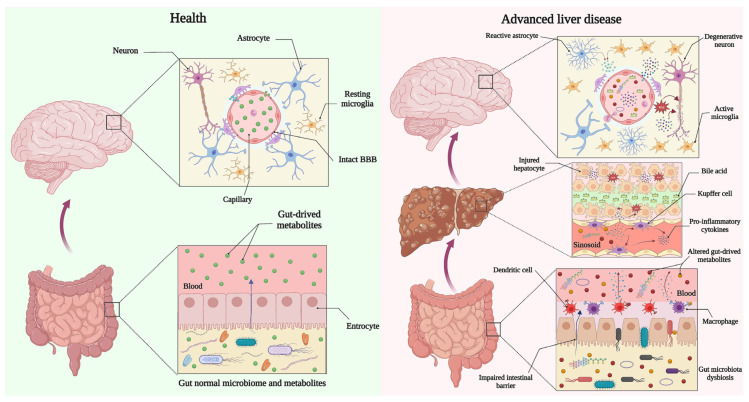
Proposed relationship between the gut microbiome and the structure of blood-brain barrier in health and advanced liver disease. Almost one to two years after birth gut microbiome community is created and its diversity is altered under the influence of age, diet, lifestyle, and geography during life. In healthy, normal gut microbiota-derived metabolites enter circulation and reach the BBB and maintain the integrity of barrier. Advanced liver disease and cirrhosis altered gut microbiota composition. Altered microbiota produces different metabolites that disrupt the intestinal barrier and result in bacterial translocation. Metabolites and products of altered bacteria (i.e., DNA, LPS, etc.) trigger the innate immune system that initiates the production of pro-inflammatory cytokines. Systemic inflammation activates sinusoidal kupffer cells and causes hepatocyte injury that along with gut-derived products and metabolites impair the integrity of the BBB. Leukocyte infiltration, glial activation, oxidative stress, neuroinflammation, and neurodegeneration are the main consequences of BBB damage. BBB: Blood-brain barrier. Created with “BioRender.com. (accessed on 11 February 2023)”.

**Table 1 biomedicines-11-01272-t001:** Alteration of gut microbial communities in HE.

Study	Groups and Gender (Women/Men)	Specimen/Samples	Microbiota Alternations		Location	References
			Decreased	Increased		
Clinical studies						
Bajaj et al., 2014	Cirrhosis with HE vs healthy control (193/64)	Multi-tagged pyrosequencing on fecal samples	Clostridiales XIV; Ruminococcaceae; Lachnospiraceae	Enterococcaeae Staphylococcaceae; Enterobacteriaceae	United States	[21]
Bajaj et al., 2015	Cirrhosis with HE and without HE vs healthy control (18/84)	Fecal specimen analysis using multi-tagged pyrosequencing techniques	Lachnospiraceae; Ruminococcaceae; Clostridiales XIV	Enterobacteriaceae; Enterococcaceae	United States	[22]
Bajaj et al., 2012	Cirrhosis with HE/cirrhosis without HE (10/50)	Sigmoid mucosal specimen using 16S ribosomal RNA (rRNA) sequencing	Roseburia	Enterococcus; Veillonella; Megasphaera; Burkholderia	United States	[23]
Zhang et al., 2013	Cirrhosis with MHE/cirrhosis without MHE (40/37)	Fecal specimen analysis using 16S rRNA-based pyrosequencing	-	Streptococcus salivarius (as a gut urease-containing bacteria)	China	[24]
Sung et al., 2019	Acute episode of OHE/compensated cirrhosis (36/129)	Profiled fecal microbiome alternations from cohort	Bacteroidetes phylum	Firmicute; Proteobacteria; Actinobacteria	Taiwan	[25]
Wang et al., 2019	Cirrhosis with MHE/ healthy controls (0/91)	16S rRNA sequencing on stool	-	Pasteurellaceae Haemophilus; Alcaligenaceae Parasutterella	China	[26]
Bajaj et al., 2012	Cirrhosis with HE/cirrhosis without HE (4/29)	Fecal specimen analysis using 16S rRNA sequencing	-	Veillonellaceae	United States	[27]
Bajaj et al., 2012	Cirrhosis with HE/healthy controls (4/29)	Fecal specimen analysis using 16S rRNA sequencing)	Clostridiales_Incertae Sedis XIV; Ruminococcaceae; Lachnospiraceae	Leuconostocaceae; Enterobacteriaceae	United States	[27]
Chen et al., 2012	Acute-on-chronic liver failure with HE/healthy controls (42/161)	Fecal microbiota analysis (16S rRNA sequencing)	Lachnospiraceae	-	China	[28]
Yukawa-Muto et al., 2022	Cirrhosis with HE/cirrhosis without HE and healthy controls (34/45)	Fecal specimen analysis using16S rRNA and metagenomic sequencing	-	Streptococcus salivarius	Japan	[29]
Hua et al., 2022	Cirrhosis with HE/cirrhosis without HE and healthy controls (13/37)	16S rRNA analysis on fecal samples	Lachnospiraceae; Turicibacterales; Turicibacter; Turicibacteraceae	Pasteurellales; Pasteurellaceae; Haemophilus; Selenomonas	China	[30]
Lin et al., 2022	Cirrhosis with MHE/ healthy controls (-)	16S rRNA high-throughput sequencing on fecal specimen	Lachnospiraceae; Roseburia; Coprpcpccus	Veillonella	China	[31]
Bajaj et al., 2021	Cirrhosis with HE/ healthy controls (0/150)	Stool metagenomics sequencing	Faecalibacterium phage; Myoviridae	-	United States	[32]
In Vivo studies						
Kang, D. J. et al., 2016	Carbon tetrachloride (CCL_4_)-induced HE/control C57BL/6 mice	Fecal samples from large intestine and cecum	Lachnospiraceae; Ruminococcaceae; Clostridiales XIV; Bifidobacteriaceae;	Staphylococcaceae; Enterobacteriaceae; Lactobacillaceae	-	[33]
Yang et al., 2022	Bile duct ligation (BDL)-induced HE/control C57BL/6 mice	Fecal samples collected from sterile cage bottom	Bacteroidetes; Bacteroidia; MB-A2-108; Erysipelotrichia; Bacteroidales; Erysipelotrichales; Muribaculaceae; Tannerellaceae; Erysipelotrichaceae; Parabacteroides; GCA-900066225 (a genus of the Lachnospiraceae family)	Firmicutes; Bacilli; Clostridiales; Clostridia; Lactobacillales; Lachnospiraceae; Alistipes; Lactobacillus murinus; Lactobacillaceae; Lachnospiraceae; Rikenellaceae; Lactobacillus	-	[34]
Đurašević et al., 2021	Thioacetamide (TAA)-induced liver injury/control rats	Fecal sample	Muribaculaceae; Desulfovibrionaceae; Lachnospiraceae	Christensenella; Rikenellaceae; Bacteroidaceae; Lactobacillaceae	-	[35]
Yang et al., 2022	CCL_4_-induced liver fibrosis/control C57BL/6 mice	Fecal sample	Staphylococcus	Bacteroides; Acinetobacter	-	[36]
Wu et al., 2022	CCL_4_-induced liver fibrosis/control C57BL/6 mice	Pyrosequencing analysis on fecal samples	Bifidobacterium; Turicibacter; (In addition to these 2 genera, 22 bacterial genera had lower abundance)	Lactobacillus; In addition to this genus, 6 bacterial genera had higher abundance	-	[37]
Cabrera-Rubio et al., 2019	BDL-induced liver injury/control C57BL/6 mice	Pyrosequencing on fecal sample	Faecalibacterium	prausnitzii; Akkermansia; Prevotella; Bacteroides; unclassified Ruminococcaceae	-	[38]
De Minicis et al., 2014	BDL-induced liver injury/control C57BL/6 mice	Fecal samples from cecum	Erysipelotrichaceae	Lachnospiraceae; Ruminococcaceae	-	[39]
Li et al., 2019	D-galactosamine (GalN)-induced ALF/control Sprague–Dawley rats	Fecal sample	Christensenellaceae; Fastidiosipila; Romboutsia	Betaproteobacteria; Burkholderiales	-	[40]

Abbreviations: ALF: Acute liver failure; BDL: Bile duct ligation; CCL_4_: Carbone tetrachloride; GalN: Galactosamine; HE: Hepatic encephalopathy; MHE: Minimal hepatic encephalopathy; OHE: Overt hepatic encephalopathy; rRNA: Ribosomal RNA; TAA: Thioacetamide.

**Table 2 biomedicines-11-01272-t002:** Evidence for disrupting the BBB in liver disease and HE.

Study	Case/Model	Method	Findings	Reference
Human studies				
Kato et al., 1992	Postmortem on 9 patients with ALF	Electron microscopic study on cerebral cortex capillaries	Swollen and vacuolated endothelial cells, intact tight junctions, enlargement and vacuolated basement membrane, vacuolated pericytes and swollen perivascular astrocyte end-feet	[90]
Sa et al., 2010	Postmortem analysis of brain samples from 14 patients with ALF	Electron microscopy	shrunken and vacuolated endothelial cells disrupted tight junctions and mitochondria, as well as swollen perivascular astrocyte end-feet	[91]
Animal studies				
Livingstone et al., 1977	devascularization and total hepatectomy-induced HE/control Wistar rats	Trypan blue, ^14^C-inulin, ^14^C-sucrose, ^14^C-glucose, ^14^C-phenylalanine, electron microscopy	Increased brain uptake index of ^14^C-substrates; raised in the cerebral concentration of trypan blue; Swollen and vacuolated perivascular astrocyte end-feet and their mitochondria	[92]
Zaki et al., 1984	Devascularized and GalN-induced ALF/control albino rats	^14^C-inulin, ^14^C-sucrose, ^14^C-glucose	Increased brain uptake index of ^14^C substrates	[93]
Mossakowski et al., 1985	TAA-induced HE rats	Electron microscopy on the cerebral cortex	Degenerative mitochondria and organelles in astrocytes,	[94]
Traber et al., 1987	GalN-induced ALF/control New Zealand White rabbits	Horseradish peroxidase injection and electron microscopy	swollen and vacuolated astrocytic foot processes, intact endothelial cells	[95]
Nguyen et al., 2006	AOM-induced ALF/control C57BL/6 mice	Evans blue dye extravasation, electron microscopy	Increased level of Evans blue dye in cerebral tissues; swollen perivascular astrocyte end-feet	[96]
Chen et al., 2009	AOM-induced ALF/control mice	Western blot analysis of tight junction proteins	Decreased cerebral proteins of occludin, claudin-5 and ZO-1	[97]
Kristiansen et al., 2010	Hepatic devascularization and portocaval anastomosis-induced ALF/control Norwegian Landrace pigs	Electron microscopy examination on frontal lobe, cerebellum, and brain stem	Perivascular edema, abnormal processes of astrocytes and pericytes, swollen neuron	[98]
Sa et al., 2010	GalN+LPS induced ALF/control BALB/c mice	Electron microscopy, immunohistochemistry, Evans blue dye extravasation	Shrunken and vacuolated endothelial cells, disrupted tight junctions, swollen perivascular astrocyte end-feet; lower protein levels of occludin; Increased cerebral level of Evans blue	[91]
Wang et al., 2011	Acetaminophen-induced ALF/control BALB/c mice	Evans blue dye extravasation, electron microscopy, western blot analysis	Increased level of Evans blue dye in brain tissues; shrunken and vacuolated endothelial cells, incomplete tight junctions, swollen perivascular astrocyte end-feet; decreased the protein expression of occludin in cerebral tissues	[99]
Quinn et al., 2014	BDL-induced liver injury/control Sprague Dawley rats	Immunofluorescence staining of brain microvasculature, Evans blue dye extravasation	losing the microvessel integrity, increased level of Evans blue dye in brain tissues;	[100]
Chastre et al., 2014	Azoxymethane (AOM)+LPS-induced ALF and coma/control C57BL/6	IgG extravasation	Increased protein expression of IgG in brain tissues	[101]
Faleiros et al., 2015	TAA-induced ALF/control C57BL/6 mice	Electron microscopy	Abnormal structure of brain capillary activated endothelial cells and disrupted tight junctions, enlargement of perivascular astrocyte end-feet	[102]
McMillin et al., 2015	AOM-induced ALF/control C57BL/6 mice	Evans blue dye extravasation	Increased cerebral level of Evans blue	[103]
Thabut et al., 2015	BDL+NH3-induced cirrhosis/control rats	Fluorochrome extravasation	Increased cerebral fluorescence intensity	[104]
Grant et al., 2018	TAA and AOM-induced HE/control C57BL/6 mice	Evans blue dye extravasation	Increased level of Evans blue dye in brain tissues	[105]

Abbreviations: ALF: Acute liver failure; AOM: Azoxymethane; BDL: Bile duct ligation; GalN: Galactosamine; IgG: Immunoglobulin G; LPS: Lipopolysaccharide; NH3: Ammonia; TAA: Thioacetamide; ZO-1: Zonula occludens-1.

**Table 3 biomedicines-11-01272-t003:** Experimental evidence of therapeutic options that target the BBB by modulating gut microbiota.

Disease (Animal Model)	Interventions	Findings	Reference
Probiotic and prebiotic			
Chronic Parkinson’s disease mouse model	Probiotic (*Lacticaseibacillus rhamnosus* GG) + prebiotic (polymannuronic acid)	Improved integrity of the BBB, prevented dopaminergic neuronal loss and increased glial cell-derived neurotrophic factor and BDNF in striatum, and inhibited striatal apoptosis	[184]
APP/PS1 mouse model of Alzheimer’s disease	Probiotic supplement (several beneficial species)	Decreased fluorescence intensity of Evans blue in hippocampus, increased expression of tight junction proteins ZO-1 and occludin in brain, and reduced concentration of LPS and pro-inflammatory cytokines in brain	[185]
Mouse model of depressive-like behavior (repeated restraint stress and lipopolysaccharide)	New probiotic agent (*Komagataella pastoris* KM71H)	Decreased extravasation of Evans blue dye in prefrontal cortex, and prevented neuroinflammation and cerebral oxidative stress	[186]
LPS-induced model of systemic inflammation in rat	Probiotic (*Lactobacillus plantarum* IS 10506)	Upregulated glial fibrillary acidic protein (GFAP) and platelet endothelial cell adhesion molecule-1 (PECAM1)in brain	[187]
Stress (chronic unpredictable mild stress and sleep deprivation in mice)	Probiotic (*Lactobacillus plantarum* MTCC 9510)	Decreased Evans blue concentration in brain, increased hippocampal BDNF, prevented neuroinflammation and oxidative stress	[188]
Lead toxicity in mice	Probiotic (*Lactobacillus fermentum* HNU312)	Increased the integrity of the BBB, decreased neuroinflammation, and improved anxiety-like and depression-like behaviors	[189]
Senescence-accelerated mouse prone 8 mouse model of aging	Probiotics (species of *Lactobacillus* and *Bifidobacterium*)	Improved disruption of the BBB, decreased astrocyte reactivity and microglial activation, reduced plasma and cerebral LPS concentrations, decreased mRNA expression of toll-like receptor 4 and nuclear factor-κB, and reduced neuroinflammation in the brain	[190]
Acute mice models of Gulf War illness	Prebiotic (andrographolide)	Restored claudin-5 protein level, increased BDNF, and decreased microglial activation in brain	[191]
Antibiotic therapy			
Transgenic MitoPark mouse model of Parkinson’s disease	Rifaximin	Decreased circulatory levels of claudin-5 and occludin (protected the BBB), suppressed systemic inflammation, reduced astrocyte reactivity, and decreased microglial activation	[192]
Mouse model of subarachnoid hemorrhage	Clarithromycin	Reduced extravasation of immunoglobulin G, and increased ZO-1 protein expression,	[193]
Mouse model of postoperative cognitive dysfunction	Cefazolin	Increased expression of ZO-1 and occludin, and decreased extravasation of Evans blue in brain tissue	[194]
Fecal microbiota transplantation			
Chronic rotenone-induced Parkinson’s disease mouse model	FMT (from control mice)	Restored tight junction proteins occludin, claudin-5, and ZO-1, decreased astrocyte reactivity, reduced microglial activation, decreased concentration of lipopolysaccharide, and suppressed neuroinflammation (TLR4/MyD88/NF-κB signaling pathway) in substantia nigra	[195]
Experimental autoimmune encephalomyelitis mouse model of multiple sclerosis	FMT (from control mice)	Decreased extravasation of Evans blue, increased protein expression of occludin-5 in the spinal cord, and reduced astrocyte reactivity, as well as decreased microglial activation in brain tissue	[196]
Mouse model of spinal cord injury	FMT (from control mice)	Reduced the level of Evans blue, restored expression of ZO-1 and occludin, decreased astrocyte reactivity, and reduced microglial activation in the spinal cord	[197]
Antibiotics-induced microbiome depletion and BBB permeabilization	FMT (pathogen-free mice)	Increased expression of ZO-1 and ZO-2 proteins in cerebral microvessels	[198]

Abbreviations: BDNF: Brain-derived neurotrophic factor; BBB: Blood-brain barrier; FMT: Fecal microbiota transplantation; GFAP: Glial fibrillary acidic protein; LPS: Lipopolysaccharide; NF-κB: Nuclear factor kappa-light-chain-enhancer of activated B cells; PECAM1: Endothelial cell adhesion molecule-1; TLR4: Toll-like receptor 4; ZO-1: Zonula occludens-1.

## Data Availability

Not applicable.

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
