# Peer review of "Gut Dysbiosis and Blood-Brain Barrier Alteration in Hepatic Encephalopathy: From Gut to Brain"

_biomedicines, 2023, doi:10.3390/biomedicines11051272_

Round 1
Reviewer 1 Report
This article is a review of the existing evidence on alterations in the microbiota in patients with encephalopathy.
Some minor aspects that I comment to the authors could increase the quality of the manuscript:
1. ALF?
2. Table 1 could be improved by incorporating other sociodemographic variables (sex, age, location...). It would also be advisable to incorporate two columns, one with the increases in the microbiota and another with the observed decreases.
3. GCA-900066225, should be specified.
4. Table 1. ref 32. 22 and 6 genera also.........?
5. line 212. HE?, delete ?
6. TAA?, AOM?
7. A table footer should be included indicating all the abbreviations contained.
8. Some references should be cited:
10.5114/aoms.2018.80651
https://doi.org/10.1111/acer.13013
Author Response
To: 20th, March, 2023
Biomedicines, Editors
Special Issue Molecular Mechanisms of Neurological Autoimmune Disorders
* We wish to express our appreciation to the reviewers for their insightful comments, which have helped us significantly to improve our manuscript. We have revised our paper accordingly and feel that your comments helped clarify and improve our paper. Please find our response (in blue) to the reviewer’s specific comments (in black) below. In the submitted revised manuscript, we have highlighted the revised text in blue color.
* Moreover, the revised version of the manuscript critically and scientifically edited by a specialized hepatologist Prof Fin Stolze Larsen from Copenhagen University Hospital.
Reviewer 1
Comments and Suggestions for Authors
This article is a review of the existing evidence on alterations in the microbiota in patients with encephalopathy.
Some minor aspects that I comment to the authors could increase the quality of the manuscript:
- ALF?
Our response: Thank for sharing. We organized all abbreviations in revised text.
- Table 1 could be improved by incorporating other sociodemographic variables (sex, age, location...). It would also be advisable to incorporate two columns, one with the increases in the microbiota and another with the observed decreases.
Our response: We considered your comments and added these items to Table 1 in revised version.
- GCA-900066225, should be specified.
Our response: We specified it in the revised manuscript.
- Table 1. ref 32. 22 and 6 genera also.........?
Our response: We revised it in the revised text.
- line 212. HE?, delete ?
Our response: We fixed it in the text.
- TAA?, AOM? 7. A table footer should be included indicating all the abbreviations contained.
Our response: Thank you for your comment. As your suggestion, we added a footer for all tables and reviewed all abbreviations in the revised version.
- Some references should be cited:
10.5114/aoms.2018.80651
https://doi.org/10.1111/acer.13013
Our response: Thank you for sharing. We cited these references in the revised text.
Sincerely yours
Dr.Ali Sepehrinezhad,
Department of Neuroscience,
Iran University of Medical Sciences, Tehran, Iran
Email: sepehrinezhad.a@iums.ac.ir

Reviewer 2 Report
Interesting review article with limited information. I have a few suggestions which may be included in this review article.
1. For the benefit of readers, the author may include enteric nervous system and it has influence on BBB. 2. The Author needs to include major gut microbiome studies (BBB, Alcohol liver disease, neuroinflammation, systemic inflammation etc.) 3. Importance of viruses (virome) needs to be included. 4. Relevance of myobiome needs to be discussed. 5. Author can discuss the sequencing method and the importance of metagenomics and metatranscriptomics. 6. Conclusion needs to be further expanded with future research direction. 7. Please include the importance of small intestine versus large intestine gut microbiome. How they differ and have a major impact on BBB disruption.Author Response
To: 20th, March, 2023
Biomedicines, Editors
Special Issue Molecular Mechanisms of Neurological Autoimmune Disorders
* We wish to express our appreciation to the reviewers for their insightful comments, which have helped us significantly to improve our manuscript. We have revised our paper accordingly and feel that your comments helped clarify and improve our paper. Please find our response (in blue) to the reviewer’s specific comments (in black) below. In the submitted revised manuscript, we have highlighted the revised text in blue color.
* Moreover, the revised version of the manuscript critically and scientifically edited by a specialized hepatologist Prof Fin Stolze Larsen from Copenhagen University Hospital.
Reviewer 2
Interesting review article with limited information. I have a few suggestions which may be included in this review article.
- For the benefit of readers, the author may include enteric nervous system and it has influence on BBB.
Our response: Thank you for your suggestion. The direct evidence for the effects of enteric nervous system modulation on the BBB has not been typically investigated in HE and gut-brain axis-affected disorders. However, we have added a section that allows readers to pay more attention to how metabolites derived from the gut microbiota modulate central nervous system functions via the enteric nervous system.
- The Author needs to include major gut microbiome studies (BBB, Alcohol liver disease, neuroinflammation, systemic inflammation etc.)
Our response: Thank you for your comment. In this review we had a special vision only on patients with hepatic encephalopathy. We presented the main gut microbiome clinical and experimental studies in hepatic encephalopathy patients in Table 1.
- Importance of viruses (virome) needs to be included.
- Relevance of mycobiome needs to be discussed.
Our response: Thank you for your suggestion. We previously considered these issues. It is unfortunate that little is known about mycobiome dysbiosis and virome composition in patients with hepatic encephalopathy. In advanced liver diseases, some studies as mentioned in the below have identified the composition of mycobiome and virome to a limited extent, but not in hepatic encephalopathy.
- Fungal dysbiosis in cirrhosis (DOI: 10.1136/gutjnl-2016-313170)
- Intestinal fungi contribute to development of alcoholic liver disease (DOI: 10.1172/JCI90562)
- The gut mycobiome: a novel player in chronic liver diseases (review; DOI: 10.1007/s00535-020-01740-5)
- Intestinal virome and therapeutic potential of bacteriophages in liver disease (review; DOI: https://doi.org/10.1016/j.jhep.2021.08.003)
- Intestinal virome in patients with alcoholic hepatitis (DOI: 10.1002/hep.31459)
There was one metagenomics study by Bajaj et al., 2021 that identified virome dysbiosis in hepatic encephalopathy patients that it was inserted to Table 1.
We also mentioned the limitations of these studies in the last part of our manuscript (i.e., Perspective and Conclusion section).
- Author can discuss the sequencing method and the importance of metagenomics and metatranscriptomics.
Our response: Thanks for sharing. We added a new part as metagenomics versus metatranscriptomics in Box 1.
- Conclusion needs to be further expanded with future research direction.
Our response: We critically revised the Conclusion section and this part was changed into Conclusion Remarks and Future Perspective section.
- Please include the importance of small intestine versus large intestine gut microbiome. How they differ and have a major impact on BBB disruption.
Our response: There are few studies examining the composition of the upper intestinal microbiota in hepatic encephalopathy, as most studies have focused on the large intestine microbiome. However, we considered your comment and compared this diversity in Box 2 of the revised manuscript.
Sincerely yours
Dr. Ali Sepehrinezhad,
Department of Neuroscience,
Iran University of Medical Sciences, Tehran, Iran
Email: sepehrinezhad.a@iums.ac.ir

Round 2
Reviewer 2 Report
It’s been revealed and suitable for publication